# Anti-Bacterial Effect of CpG-DNA Involves Enhancement of the Complement Systems

**DOI:** 10.3390/ijms20143397

**Published:** 2019-07-10

**Authors:** Te Ha Kim, Joongwon Park, Dongbum Kim, Avishekh Gautam, Madhav Akauliya, Jinsoo Kim, Hanseul Lee, Sangkyu Park, Younghee Lee, Hyung-Joo Kwon

**Affiliations:** 1Department of Microbiology, College of Medicine, Hallym University, Chuncheon 24252, Korea; 2Center for Medical Science Research, College of Medicine, Hallym University, Chuncheon 24252, Korea; 3Department of Biochemistry, College of Natural Sciences, Chungbuk National University, Cheongju 28644, Korea

**Keywords:** cobra venom factor, complement, CpG-DNA, *E. coli* K1, infection

## Abstract

CpG-DNA activates the host immune system to resist bacterial infections. In this study, we examined the protective effect of CpG-DNA in mice against *Escherichia coli* (*E. coli*) K1 infection. Administration of CpG-DNA increased the survival of mice after *E. coli* K1 infection, which reduces the numbers of bacteria in the organs. Pre-injection of mice with CpG-DNA before *E. coli* K1 infection increased the levels of the complement C3 but not C3a and C3b. The survival of the mice after *E. coli* K1 infection was significantly decreased when the mice were pre-injected with the cobra venom factor (CVF) removing the complement compared to the non-CVF-treated mice group. It suggests that the complement has protective roles against *E. coli* K1 infection. In addition, the survival of complement-depleted mice was increased by CpG-DNA pre-administration before *E. coli* K1 infection. Therefore, we suggest that CpG-DNA enhances the anti-bacterial activity of the immune system by augmenting the levels of complement systems after *E. coli* K1 infection and triggering other factors as well. Further studies are required to investigate the functional roles of the CpG-DNA-induced complement regulation and other factors against urgent bacterial infection.

## 1. Introduction

The host protection by the innate immune system against pathogens is based on the recognition and detection of many types of pathogenic products [1]. Toll-like receptor 9 (TLR9), one of the pattern recognition receptors, distinguishes bacterial DNA from self-DNA, and TLR9-bacterial DNA interaction promotes immunomodulatory effects in the host [2]. Bacterial DNA and synthetic CpG-oligodeoxynucleotides (CpG-DNA) induce the differentiation and proliferation of immune cells [2,3,4], regulate production of Th1-related cytokines [5,6], and complement activation depending on the TLR9 [7]. TLR9 has a resistance effect against *Mycobacterium tuberculosis* in cooperation with TLR2 [8]. It was reported that TLR9 contributes to improved protective innate immunity against Gram-negative bacteria pneumonia by rapid accumulation of dendritic cells in the lung [9]. TLR9 also leads to bacterial clearance against *Acinetobacter baumannii* and methicillin-resistant *Staphylococcus aureus* (MRSA) infection [10,11].

CpG-DNA treatment stimulates the innate immunity including the activation of dendritic cells and the induction of oxidative stress against the infections of several bacteria such as *Listeria monocytogenes* [12], *Staphylococcus aureus* (*S. aureus*) [13], *Salmonella typhimurium* [14], and *Klebsiella pneumonia* [15]. Furthermore, systemic administration of CpG-DNA prevents intracerebral *Escherichia coli* (*E. coli*) K1 infection by increasing the monocytes in brain and spleen of neutropenic mouse [16]. In recent studies, we proved the protective effects of the antibodies produced by CpG-DNA administration against MRSA, which is one of the Gram-positive bacteria, infection [17,18].

Although most *E. coli* strains are not harmful, some virulent strains have detrimental effects on the host, which causes a variety of diseases such as diarrhea, urinary tract infections, sepsis, meningitis, and peritonitis [19]. Antibiotics have been used to treat *E. coli*-induced diseases, but it has been complicated because of the emergence of multidrug-resistant *E. coli* [20,21,22]. Because the rate of *E. coli* resistance to antibiotics is increasing, other strategies for the treatment of *E. coli* infections are urgently required. The development of vaccines with mixtures of live attenuated or inactivated strains has been studied to protect against *E. coli* infection [23,24]. As well as cellular vaccines, several adhesion proteins and toxins from *E. coli* were used to construct a multi-epitope fusion antigen to produce antibodies neutralizing several adhesion proteins and toxins from *E. coli* [25].

In this case, we investigated the anti-bacterial roles of CpG-DNA against *E. coli* K1 infection. Administration of CpG-DNA in the mouse peritoneal cavity activated the complement and enhanced the anti-bacterial effects against *E. coli* K1 infection. This novel function of CpG-DNA provides more understanding into the anti-bacterial effects of CpG-DNA and suggests that investigation of the CpG-DNA-induced complement regulation can be helpful for the treatment of *E. coli* infection.

## 2. Results

### 2.1. Intraperitoneal Administration of CpG-DNA Protects Mice against E. coli K1 Infection

To investigate the protective role of CpG-DNA against *E. coli* K1 infection, experiments were performed in a mouse infection model using BALB/c mice, as depicted in Figure 1a. The lethal dose of intraperitoneal *E. coli* K1 infection in BALB/c mice was determined as 1 × 10^6^ colony forming units (CFU)/mouse (Appendix A), which resulted in the death of the mice within 48 h. The BALB/c mice were intraperitoneally (i.p.) injected with PBS or CpG-DNA 1826 (2.5 mg/kg mouse). After seven days, the mice were i.p. injected with PBS or *E. coli* K1 (1 × 10^6^ CFU/mouse), and the survival rates were observed for 48 h. All of the mice pre-treated with CpG-DNA before *E. coli* K1 infection survived, but all of the mice that only received *E. coli* K1 died within 48 h (Figure 1b). One day after *E. coli* K1 infection, the liver, lungs, kidneys, spleen, blood, and peritoneal fluids were extracted from the mice to measure the bacterial burden. Bacterial infections were detected in all the examined samples with the highest CFU in the kidneys, but the bacterial loads in all of the examined samples except for the blood were decreased in the mice pre-treated with CpG-DNA 1826 prior to infection (Figure 1c). The histopathology of the tissues was also inspected one day after *E. coli* K1 infection. Signs of abnormality were detected only in the kidneys after the *E. coli* K1 infection, but severe histopathological damages were not observed in the group pre-treated with the CpG-DNA 1826 before infection (Figure 1d). Taken together, these results suggest that pre-treatment with CpG-DNA in the peritoneal cavity enhances survival and bacterial clearance in mice after peritoneal *E. coli* K1 infection.

### 2.2. Pre-Treatment of CpG-DNA Induces a Significant Recovery in the Mice after E. coli K1 Infection

To investigate the effect of CpG-DNA treatment prior to *E. coli* K1 infection on the recovery of the infected mice, additional experiments were performed according to the procedure depicted in Figure 2a. Body weight was observed for seven days after intraperitoneal *E. coli* K1 infection. The body weight of the *E. coli* K1-infected mice was significantly decreased after the infection, and all the mice died within 48 h. However, the body weights of the mice pre-treated with CpG-DNA prior to *E. coli* K1 infection were decreased and recovered within seven days (Figure 2b). After seven days, the liver, lungs, kidneys, spleen, blood, and peritoneal fluids were extracted from the mice, and the bacterial loads were assessed. No bacterial infection was detected in the examined samples after seven days of the infection (Figure 2c). The histopathology was also determined in the indicated organs, and no severe histopathological damages were detected in the organs (Figure 2d). Therefore, the results suggest that pre-treatment with CpG-DNA induced a significant recovery in the mice after peritoneal *E. coli* K1 infection.

### 2.3. Complement Activation by CpG-DNA Administration and/or E. coli K1 Infection

It was reported that CpG-DNA affects complement activation, which results in the modulation of the immune system by upregulation of CD40 and CD83 and the induction of cytokines (IL-6 and TNF-α) in the whole blood loop system [7]. To determine the levels of the complement components in the mouse serum after the administration of CpG-DNA, BALB/c mice were i.p. injected with CpG-DNA 1826. The sera were obtained from the mice at the indicated time points, and the levels of the serum complement components were measured using ELISA kits. The levels of C3 were increased one day after the CpG-DNA injection and decreased after three days, and the levels of C3a and C3b increased up to five days after the injection and then decreased to similar levels of the PBS-injected control mice seven days after the injection (Figure 3a). To determine the change in the levels of complement components after CpG-DNA 1826 and/or *E. coli* K1 infection, BALB/c mice were i.p. injected with CpG-DNA. Seven days later, the mice were i.p. injected with *E. coli* K1. After one more day, the sera were collected from the mice, and the levels of C3, C3a, and C3b in the serum were assessed using ELISA kits. *E. coli* K1 infection induced a drastic increase in the C3a and C3b levels when compared to the PBS group. However, there was no change in the C3 levels after *E. coli* K1 infection (Figure 3b). *E. coli* K1 infection can activate the complement system and increase C3a and C3b through the classical pathway, the alternative pathway, and the lectin pathway. Therefore, the reason why the amount of C3 did not reduce greatly is presumed to be the induction of C3 in response to *E. coli* K1 infection. In the mice pre-treated with CpG-DNA prior to *E. coli* K1 infection, the C3 levels were significantly increased, but the levels of C3a and C3b were similar to those of the control mice (Figure 3b). Therefore, it is likely that the increase of the C3 level is related a reduced cleavage of C3 to C3a/C3b due to the CpG-DNA treatment. It suggests that the regulation of these complement levels might contribute to the survival of the mice against *E. coli* K1 infection.

### 2.4. Depletion of C3 Causes a High Mortality Rate in the Mice against E. coli K1 Infection

To determine the protective roles of C3 for the survival of the *E. coli* K1-infected mice, we used the C3-depleting cobra venom factor (CVF). When BALB/c mice were i.p. injected with CpG-DNA 1826 and then treated with CVF, C3 in the sera was depleted both in the PBS and CpG-DNA pre-treated mice (Figure 4a,b). To determine the CFU of *E. coli* K1 leading to the death of the mice after CVF injection, BALB/c mice were i.p. injected with CVF and with the indicated CFU of *E. coli* K1, and the survival rates were monitored for 30 h (Appendix A). We selected 2.5 × 10^5^ CFU as an optimal CFU because the dose induced the most moderate survival rate in a time-dependent manner. To investigate the roles of the increased C3 for the survival of the *E. coli* K1-infected mice, BALB/c mice were i.p. injected with PBS or CpG-DNA 1826, treated with PBS or CVF, and i.p. injected with PBS or *E. coli* K1 (2.5 × 10^5^ CFU), and the survival of the mice was observed for 48 h (Figure 4c). After the *E. coli* K1 infection, CVF-treated mice died within 33 h, but most of the PBS control mice (90%) survived for 48 h (Figure 4d). This result suggests that the levels of C3 are very important for the protection of mouse models against *E. coli* K1 infection. When the mice were pre-treated with CpG-DNA, the survival of the mice after *E. coli* K1 infection was higher than that of the PBS-injected mice (100% vs. 90% at 48 h, Figure 4d). When the mice were subjected to CVF injection and *E. coli* K1 infection, pretreatment with CpG-DNA enhanced the survival rate compared to the PBS-treatment (20% at 48 h vs. 0% at 33 h). These results suggest that other CpG-DNA-induced regulators, besides the complement components, might be involved in the protection of the host against *E. coli* K1 infection.

### 2.5. Phagocytosis Is Not Increased by CpG-DNA Administration in the Mice Peritoneal Cavity against E. coli K1 Infection

In addition to complement components, other factors exist for host protection against bacterial infection. We supposed that antibodies-produced by CpG-DNA could be the one to resist *E. coli* K1 infection based on our previous studies [17]. To determine the production of *E. coli* K1-reactive antibodies by CpG-DNA administration in the mouse peritoneal cavity and serum, BALB/c mice were i.p. injected with PBS or CpG-DNA 1826. After seven days, peritoneal fluid and sera were extracted from the mice, and amounts of the IgG subclasses were analyzed by ELISA. The amounts of *E. coli* K1-reactive antibodies were increased by CpG-DNA administration both in the peritoneal fluid and sera (Figure 5a,b). These results suggest that CpG-DNA produces *E. coli* K1-reactive antibodies in the mouse peritoneal cavity and sera. We used CpG-DNA-induced monoclonal antibody (3F5H6 mIgG) obtained from the previous studies [17] to apply *E. coli* K1-infected mouse models. We analyzed the effect of 3F5H6 mIgG on phagocytic capacity in the *E. coli* K1-infected mouse. Fluorescein isothiocyanate (FITC)-labeled *E. coli* K1 was incubated with the PBS, normal mouse IgG (normal mIgG), or 3F5H6 mIgG and then injected i.p. in mice. We analyzed the phagocytic level in mouse peritoneal cavity macrophages by flow cytometry. In contrast to the results in *S. aureus* MW2 [17], phagocytic index against *E. coli* K1 decreased in the presence of the 3F5H6 mIgG (Figure 5c).

To examine the effect of CpG-DNA on phagocytosis in the mouse peritoneal cavity, CpG-DNA was administered in the mouse peritoneal cavity and the FITC-labeled *E. coli* K1 was injected i.p. in mice and then the phagocytic level was analyzed in the mouse peritoneal cavity by flow cytometry. Phagocytic activity of macrophages against *E. coli* K1 was not increased by CpG-DNA in mouse peritoneal cavity (Figure 5d). These results suggest that CpG-DNA and 3F5H6 mIgG do not enhance the phagocytic activity against *E. coli* K1 infection.

## 3. Discussion

CpG-DNA enhances the activity of the host immune system against intracellular and extracellular bacterial species [15,26]. Therefore, administration of CpG-DNA in in vitro and in vivo systems protects the host against various Gram-positive and Gram-negative bacterial infections. In this case, we focused on the protective functions of the CpG-DNA-induced complement components in mouse models against *E. coli* K1 infection.

Several studies have suggested that pre-administration of CpG-DNA within three days before bacterial infection leads to the expression of several cytokines such as IL-6, IL-12, IFN-γ, and TNF-α, which enhances the host defense system [5,15,27]. Furthermore, we investigated factors related to the anti-bacterial effects of CpG-DNA administration against MRSA infection, as described in a previous report [17,18]. We confirmed in this study that CpG-DNA administration significantly increased the level of complement C3 and enhanced the survival of mice and the bacterial clearance in the examined main organs after *E. coli* K1 infection. CpG-DNA activates the complement system in the whole blood loop system [7]. Production of complement components are produced by hepatocytes and endothelial and epithelial cells and by immune cells such as monocytes, macrophages, and dendritic cells [28]. In this study, we used phosphorothioate-modified CpG-DNA 1826. It was previously reported that phosphorothioate-modified oligonucleotides reduced levels of factor H, which is an inhibitor of the alternative pathway convertase, and affected alternative pathway, which leads to complement activation [29,30]. It was also reported that CpG-DNA increases the levels of C3a and C5a [7]. In this case, we found that injection of CpG-DNA1826 into mice induced a transient increase of C3 followed by later and transient increase of C3a and C3b. Several reports have suggested that uncontrolled complement activation, production of C3a and C5a, by bacterial infections leads to multiple organ failure and death [31,32,33]. We investigated the changes in the CpG-DNA-induced complement component (C3, C3a, and C3b) levels in mice and found that the amount of the complement components were transiently increased by the administration of CpG-DNA 1826 compared to the non-treated group. Seven days after CpG-DNA 1826 injection, the amount of the complement components returned to their normal state (Figure 3a). Therefore, the results suggest that the potentially harmful effects of the CpG-DNA-induced complement components also could have been minimized when we injected *E. coli* K1 into the mice. The amounts of C3a and C3b were increased in the serum of mice after *E. coli* K1 infection, as we expected (Figure 3b), and the mice died within 48 h after *E. coli* K1 infection (Figure 1). Therefore, the high increase of C3a and C3b after the infection might be one of the reasons leading to the death of the mice early after *E. coli* K1 infection. When the mice were pre-treated with CpG-DNA prior to *E. coli* K1 infection, the C3 levels were significantly increased, but the levels of C3a and C3b did not change (Figure 3b). Considering that the CpG-DNA-pre-treated mice did not die after *E. coli* K1 infection, we suppose that the CpG-DNA-induced increase of C3 levels may have protective roles against *E. coli* K1 infection and that the prevention of the drastic increase in C3a and C3b by the CpG-DNA treatment may contribute to enhanced survival. Considering that there is a report that ssDNA and dsDNA can bind with C1q and affect the classical pathway [34], the function of other complements such as C1q, C2, and C4 can be involved. In this study, we did not check other complements, but focused on the fact that CpG-DNA is involved in complement activation and contributes to the suppression of *E. coli* K infection. Detailed effects of CpG-DNA on the complement system are to be determined in the future.

When the mice were C3-depleted using CVF before the infection, the survival of the CVF-treated mice was significantly decreased against *E. coli* K1 infection compared to the non-CVF-treated mice (Figure 4d). It supports our hypothesis that C3 has protective functions against *E. coli* K1 infection. Pretreatment with CpG-DNA increased the survival of the CVF-treated mice (Figure 4d). Therefore, we believe that other factors in addition to complement components were involved in the protective effects of CpG-DNA against *E. coli* K1 infection.

Previously, we investigated that the administration of CpG-DNA increased survival of mice against infection with MRSA and facilitated bacterial clearance in mice [17]. CpG-DNA protected immune cell populations against MRSA infection in the peritoneal cavity, bone marrow, and spleen. Injection of mice with CpG-DNA increased bacteria-reactive antibodies in the peritoneal fluid and serum depending on TLR9. Stimulation of peritoneal B cells with CpG-DNA induced bacteria-reactive antibodies in vitro. The bacteria-reactive antibodies were produced in both B1 and B2 cells of peritoneal cavity in response to CpG-DNA and the antibodies enhanced phagocytic activity of peritoneal cells. We established bacteria-reactive monoclonal antibody (3F5H6 mIgG) using CpG-DNA stimulated-peritoneal B cells. The monoclonal antibody enhanced phagocytosis. In this study, we observed antibodies reactive to *E. coli* K1 in the peritoneal cavity and serum of CpG-DNA-stimulated mice (Figure 5a,b). Apart from MRSA, CpG-DNA-induced bacteria-reactive monoclonal antibody (3F5H6 mIgG) did not increase but did decrease phagocytosis of *E. coli* K1 (Figure 5c). This suggests a possibility that 3F5H6 mIgG binds to *E. coli* K1 and inhibits the uptake of macrophage. Injection of CpG-DNA did not increase uptake of *E. coli* K1 in the peritoneal macrophages either (Figure 5c). While the peritoneal cavity macrophage of BALB/c mice showed phagocytosis of MRSA at 20% efficiency [17], phagocytosis of *E. coli* K1 was found at about 60% efficiency in the absence of CpG-DNA treatment, which suggests the presence of the mechanism inducing more efficient removal of gram-negative bacteria in the mice.

Previously, it was reported that the E3 ubiquitin ligase tripartite motif protein 29 (TRIM29) could be induced by dsDNA [35]. TRIM29 could negatively regulate the infection of gram negative bacteria *Haemophilus influenza* by targeting NEMO and played a protective role in the host defense against bacterial infection [36]. Therefore, we can consider possible involvement of TRIM29 in the protective function of CpG-DNA against *E. coli* K. However, it is not likely because expression of TRIM29 was not induced by CpG-DNA treatment [37].

Considering that several types of immune cell populations such as B cells, monocytes, macrophages, and dendritic cells were activated after CpG-DNA administration [17], we suppose that these CpG-DNA-activated immune cells and immunoregulators including complement components are involved in the enhanced survival of the mice and bacterial clearance in the examined organs after *E. coli* K1 infection. Accordingly, future studies should be concentrated on the functional role of the CpG-DNA-induced complement regulation and other factors that fight bacterial infections, which might be valuable for the treatment of infectious diseases.

## 4. Materials and Methods

### 4.1. CpG-DNA

CpG-DNA 1826 (5′-TCCATGACGTTCCTGACGTT-3′) was prepared by GenoTech (Daejeon, Korea). The backbones of the sequences were adjusted with phosphorothioate. The oligodeoxynucleotides were suspended in distilled water, and the amounts of endotoxin (>1 ng/mg of CpG-DNA 1826) were identified by a Limulus amebocyte assay (Whittaker Bioproducts, Walkersville, MD, USA).

### 4.2. Mouse

Eight-week-old BABL/c mice (Nara Biotech, Inc., Seoul, Korea) were maintained in a Biosafety level 2 facility (20–25 °C, 40–45% humidity, 12-h light/dark cycle, food and water access, *ad libitum*) in the Hallym Clinical and Translational Science Institute. Mice were anesthetized with 3% to 5% isoflurane (Pharmaceutical, Seoul, Korea) inhalation to minimize any torment. The mice were sacrificed by CO_2_ inhalation after the experiments, and all efforts were made to limit suffering. All mouse experiments were carried out following the Guide for the Care and Use of Laboratory Animals of the National Veterinary Research and Quarantine Service of Korea with the approval of the Institutional Animal Care and Use Committee of Hallym University (Permit Number: Hallym2017-63, 16 January 2018; Hallym2018-24, 12 September 2018).

### 4.3. Bacteria and In Vivo Infection Experiments

*Escherichia coli* (*E. coli* K1, serovar O1:K1:H7, KCCM 12119) were obtained from the Korean Culture Center of Microorganisms (KCCM, Seoul, Korea). *E. coli* K1 was cultured at 37 °C in Lysogeny broth (LB) overnight and re-cultured in fresh media at a 1/50 dilution until the mid-log phase (OD_600_ 0.5–0.6), and the harvested. *E. coli* K1 was washed with PBS, centrifuged, and diluted to 5 × 10^6^ colony forming units (CFU)/mL in PBS. BABL/c mice were injected with PBS or CpG-DNA 1826 (2.5 mg/kg mouse) intraperitoneally (i.p.). Seven days after the injection, the mice were injected i.p. with 0.2 mL of PBS or the bacterial suspension. Following infection, the mice were monitored for morbidity or recovery for 2 days. We investigated the survival rate, histopathology, and bacterial loads (CFU) in the control and infected mouse organs. We measured the body weight for 7 days when necessary.

### 4.4. Hematoxylin and Eosin (H&E) Stain

Paraffin embedding and tissue sectioning were performed as described previously [17]. After infection with *E. coli* K1, the liver, lungs, kidneys, and spleen were isolated and fixed in 4% buffered formalin solution and embedded in paraffin. The organs were cut into 4-μm thick sections on slides, and the tissue slides were incubated at 60 °C for 30 min, to dissolve the paraffin. After re-hydration in xylene through a progression of 100–70% ethanol, the tissues were then stained with Gill’s Hematoxylin V (Muto Pure Chemicals, Tokyo, Japan) and Eosin Y solution (Sigma-Aldrich, St. Louis, MO, USA). The tissues were dehydrated in a progression of 70–100% ethanol, incubated in xylene, and mounted with Malinol medium (Muto Pure Chemicals, Tokyo, Japan). Images of the stained tissues were scanned with an Eclipse E200 microscope (Nikon, Tokyo, Japan).

### 4.5. Colony Forming Units in the Organs

Each organ was collected from the mice and weighed. The organs were homogenized in PBS with stainless steel beads (Qiagen, Hilden, Germany). The homogenized samples were transferred to 6-well plates containing LB-Bacto™ Agar and cultured overnight at 37 °C, and the number of colonies were counted.

### 4.6. Measurement of the Complement Components

BALB/c mice were i.p. injected with PBS or CpG-DNA 1826 (2.5 mg/kg mouse). At the indicated time points, sera were collected from the mice using the cardiac puncture method. The levels of C3 were measured using the Mouse C3 ELISA Kit (Abcam, Catalogue No. ab157711, Cambridge, England) and the levels of C3a and C3b were measured using the Mouse C3a ELISA Kit (MyBioSource, San Diego, CA, USA, Catalogue No. MBS268280) and Mouse C3b ELISA Kit (MyBioSource, Catalogue No. MBS269569). The Mouse C3 ELISA Kits did not have cross-reactivity to the C3a and C3b standard. The Mouse C3a and Mouse C3b ELISA Kits did not react with the C3 standard, either. Seven days after the PBS or CpG-DNA 1826 (2.5 mg/kg mouse) injection (i.p.), the mice were i.p. injected with PBS or *E. coli* K1 (1 × 10^6^ CFU/mouse). One day after the infection, sera were obtained from the mice, and the levels of C3, C3a, and C3b were measured using the ELISA kits.

### 4.7. C3 Depletion and In Vivo Experiments

CVF was purchased from Quidel (San Diego, CA, USA) [38]. BALB/c mice were i.p. injected with PBS or CpG-DNA 1826 (2.5 mg/kg mouse). After 7 days, the mice were i.p. injected with PBS or CVF 30 μg/mouse. After 6 h, the sera were collected from the mice, and the levels of C3 were measured using the Mouse C3 ELISA Kit (Abcam). BALB/c mice were i.p. injected with PBS or CpG-DNA 1826. After 7 days, the mice were i.p. injected with PBS or CVF 30 μg. After 6 h, the mice were i.p. injected with 2.5 × 10^5^ CFU of *E. coli* K1. Survival was monitored for 2 days.

### 4.8. ELISA

To determine the production of bacteria-reactive antibodies following CpG-DNA 1826 injection in mouse peritoneal cavity, poly-L-lysine coated plates (Corning Inc, Corning city, NY, USA) were utilized as described previously [17]. *E. coli* K1 were covered with *E. coli* K1 (5 × 10^6^ CFU/well) in ELISA coating buffer (15 mM Na_2_CO_3_, 35 mM NaHCO_3_, pH 9.6) and incubated overnight at 4 °C. After incubation, the wells were fixed with 0.5% glutaraldehyde in PBS for 15 min at room temperature. After washing twice with PBS, each well was incubated with RPMI 1640 medium containing 100 mM glycine and 0.1% BSA for 30 min at room temperature to neutralize glutaraldehyde. The *E. coli* K1-coated wells were then blocked with PBS containing 1% BSA for 1 h at room temperature. Peritoneal cavity fluid and serum from PBS- or CpG-DNA 1826-i.p. injected mice were sequentially diluted, added to each well, and incubated for 1 h at room temperature. The wells were washed three times with PBS-T (0.2% Tween-20 in PBS) and antibodies, including horseradish peroxidase (HRP)-labeled goat anti-mouse IgG, IgG1, IgG2a, IgG2b, or IgG3 (Southern Biotech, Birmingham, AL, USA), which were added to the wells and incubated for 1 h at room temperature. After washing with PBS-T four times, the TMB Microwell Peroxidase Substrate Kit (KPL, Gaithersburg, MD, USA) was utilized to recognize peroxidase-labeled conjugates (blue-color expression), which was followed by the addition of TMB Stop solution (KPL) (yellow-color). The absorbance was measured at 450 nm using a SpectraMax 250 microplate reader (Molecular Devices, Sunnyvale, CA, USA).

### 4.9. Analysis of E. coli K1 Uptake in the Mouse Peritoneal Cavity

*E. coli* K1 was labeled with fluorescein isothiocyanate (FITC, Sigma-Aldrich) in 0.1 M Na_2_CO_3_ buffer (pH 8.5), as described previously [17]. To analyze the effect of bacteria-reactive monoclonal antibody (3F5H6 mIgG) on phagocytosis, FITC-labeled *E. coli* K1 cells (1 × 10^6^ CFU/mL) were incubated with PBS, normal mouse IgG, or 3F5H6 mIg (10 μg/mL) for 1 h at 4 °C and then 200 μL of the mixture i.p. injected into BALB/c mice (*n* = 3). After 90 min, peritoneal cells were harvested from the mice and stained with specific markers. To analyze the effect of CpG-DNA on phagocytosis, BALB/c mice were i.p. injected with PBS or CpG-DNA 1826 (2.5 mg/kg mouse). After 7 days, the mice were i.p. injected with PBS or FITC-labeled *E. coli* K1 (2 × 10^6^ CFU/mouse). After 90 min, peritoneal cells were harvested, counted, BALB/c mice were i.p. injected with PBS or CpG-DNA 1826 (2.5 mg/kg mouse). After 7 days, the mice were i.p. injected with PBS or FITC-labelled *E. coli* K1. After 90 min, peritoneal cells were collected from the mice and centrifuged at 1200 rpm for 5 min for analysis of FITC-labeled *E. coli* K1 uptake. The peritoneal cells were resuspended in RPMI 1640 medium and then washed with FACS buffer (1% FBS in PBS). To block the Fc receptors, the cells were incubated with 10 µg/mL anti-FcγRII/III antibody (Catalogue No: 553142, BD Biosciences, San Jose, CA, USA) for 20 min at 4 °C. The peritoneal cells were pre-gated (FSC^low^SSC^high^) for the enriched myeloid population. For the macrophages population, the antibodies used were PE-conjugated anti-CD11c (Catalogue No: 557401, BD Biosciences, San Jose, CA, USA), APC-conjugated anti-CD11b (Catalogue No: 553312, BD Biosciences), APC-eF780-conjugated anti-Gr-1 (Catalogue No: 47-5931-82, eBioscience), eF450-conjugated anti-F4/80 (Catalogue No: 48-4801-80, eBioscience), and PerCP-Cy5.5-conjugated anti-MHC II (Catalogue No: 46-5321-82, eBioscience). For the M1 macrophages population, the antibodies used were PE-conjugated anti-iNOS (Catalogue No: 12-5920-80, eBioscience, San Diego, CA, USA), APC-conjugated anti-CD206 (Catalogue No: 17-2061-80, eBioscience), APC-eF780-conjugated anti-Gr-1 (Catalogue No: 47-5931-82, eBioscience), eF450-conjugated anti-F4/80 (Catalogue No: 48-4801-80, eBioscience), and PerCP-Cy5.5-conjugated anti-MHC II (Catalogue No: 46-5321-82, eBioscience). After 1 h of incubation at 4 °C, the cells were washed with FACS buffer and the uptake of *E. coli* K1 by M1 macrophages was analyzed with a FACSCanto^TM^ II (Becton Dickinson, Franklin Lakes, NJ, USA).

### 4.10. Statistical Analysis

Data are presented as the mean ± standard deviation. Differences between two samples were evaluated by a Student’s *t*-test, and a resulting value of *P* < 0.05 was considered statistically significant.

## Figures and Tables

**Figure 1 ijms-20-03397-f001:**
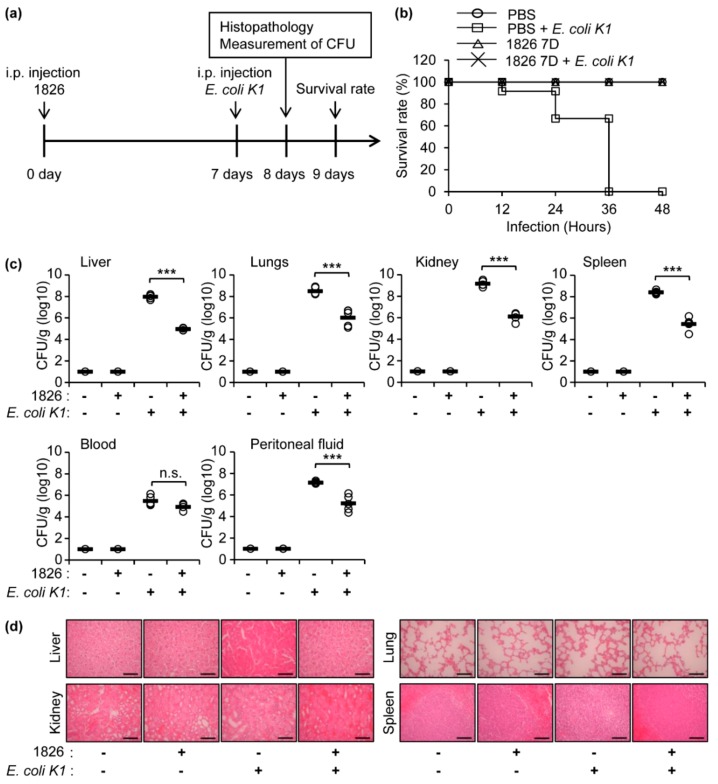
Protective effects of CpG-DNA in mice from peritoneal *E. coli* K1 infection. (**a**) Schematic diagram of the experimental process is shown. BALB/c mice were i.p. injected with PBS or CpG-DNA 1826 (2.5 mg/kg mouse). After seven days, the mice were i.p. injected with PBS or *E. coli* K1 (1 × 10^6^ CFU/mouse). (**b**) Survival of the mice was monitored for 48 h after *E. coli* K1 infection. The rate of surviving mice in each group is depicted (*n* = 12/group). (**c**) One day after *E. coli* K1 infection, the indicated organs, blood, and peritoneal fluids were isolated and homogenized in PBS (*n* = 5/group). The homogenates were serially diluted and loaded onto agar plates to measure the *E. coli* K1 CFU. (**d**) Histopathology of the indicated organs one day after the infection. Scale bar, 10 μm. 1826, CpG-DNA 1826. D, days. *** *p* < 0.0005. n.s., not significant.

**Figure 2 ijms-20-03397-f002:**
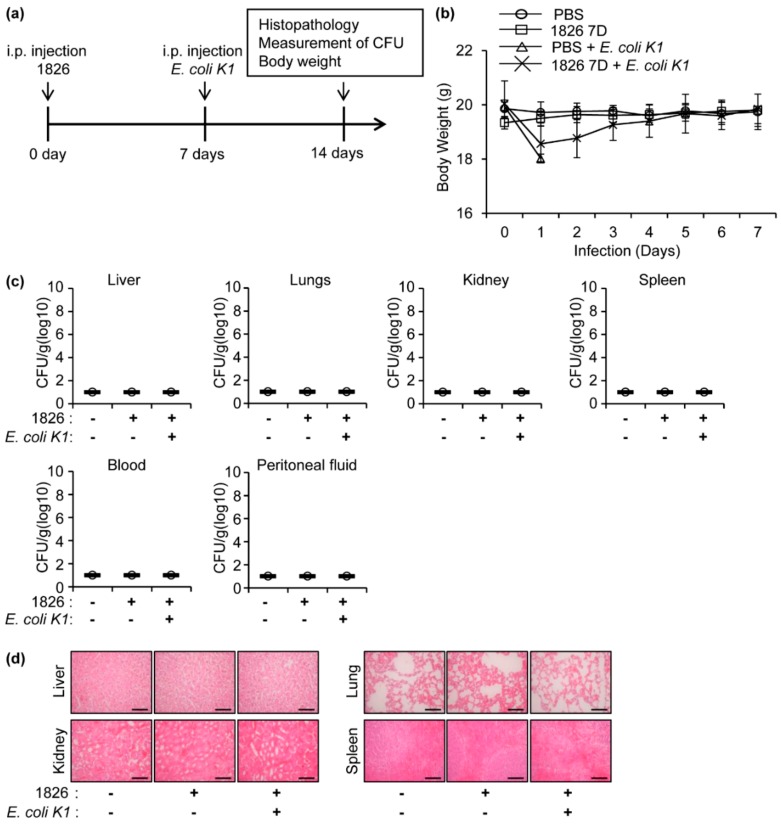
The enhanced recovery of the *E. coli* K1-infected mice by CpG-DNA pre-administration. (**a**) Schematic procedure of the experiments. BALB/c mice were i.p. injected with PBS or CpG-DNA 1826 (2.5 mg/kg mouse), followed by the i.p. injection of PBS or *E. coli* K1 (1 × 10^6^ CFU). (**b**) The body weight of the mice was monitored for seven days after *E. coli* K1 infection (*n* = 5/group). (**c**) Seven days after *E. coli* K1 infection, the surviving mice were sacrificed, and the indicated organs were isolated and homogenized in PBS. The homogenates (*n* = 5/group) were serially diluted, and loaded onto agar plates to measure the *E. coli* K1 CFU. (**d**) Histopathology of the organs at seven days after *E. coli* K1 infection. Scale bar, 10 μm. 1826, CpG-DNA 1826. D, days.

**Figure 3 ijms-20-03397-f003:**
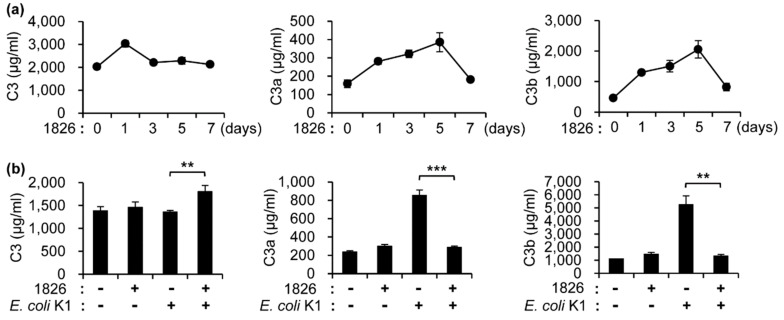
Changes in the complement components after CpG-DNA 1826 and/or *E. coli* K1 injections. (**a**) BALB/c mice were i.p. injected with CpG-DNA 1826 (2.5 mg/kg mouse). At the indicated time points, the sera were harvested from the mice using the cardiac puncture method, and the levels of the complement components (C3, C3a, and C3b) in the serum were measured using the ELISA kit (*n* = 3/group). (**b**) BALB/c mice were i.p. injected with PBS or CpG-DNA 1826. After seven days, the mice were i.p. injected with PBS or *E. coli* K1 (1 × 10^6^ CFU), and the sera were harvested from the mice using the cardiac puncture method one day after *E. coli* K1 infection. The levels of the complement components (C3, C3a, and C3b) were measured using ELISA kits (*n* = 3/group). 1826, CpG-DNA 1826. *** p* < 0.005, *** *p* < 0.0005.

**Figure 4 ijms-20-03397-f004:**
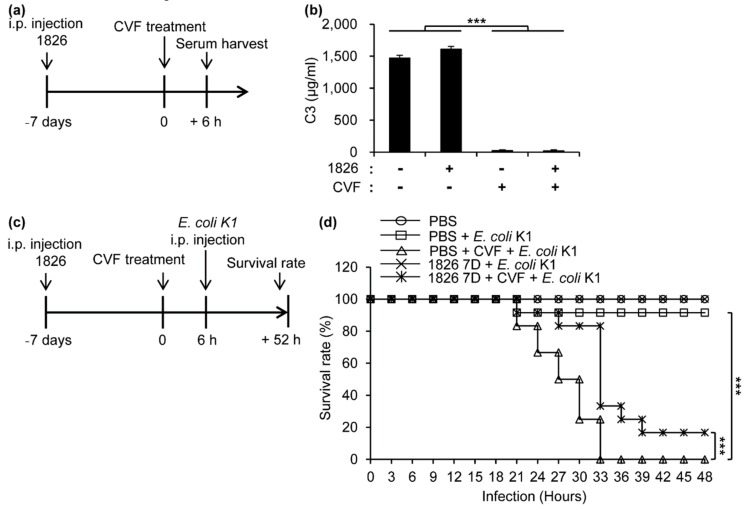
The protective role of complement 3 (C3) induced by CpG-DNA 1826 against *E. coli* K1 infection. (**a**,**b**) BALB/c mice were i.p. injected with PBS or CpG-DNA 1826 (2.5 mg/kg mouse). Seven days after the injection, PBS or cobra venom factor (CVF, 30 μg/mouse) was i.p. injected to remove the C3. After 6 h, the sera were harvested from the mice. (**b**) The levels of C3 in the serum were measured using ELISA kit (*n* = 3/group). (**c**,**d**) BALB/c mice were i.p. injected with PBS or CpG-DNA 1826. Seven days after the injection, PBS or cobra venom factor (CVF, 30 μg/mouse) was i.p. injected to remove C3. After 6 h, the mice were i.p. injected with PBS or *E. coli* K1 (2.5 × 10^5^ CFU). (**d**) The survival was monitored for another 48 h after *E. coli* K1 infection (*n* = 12/group). 1826, CpG-DNA 1826. CVF, cobra venom factor. D, days. *** *p* < 0.0005.

**Figure 5 ijms-20-03397-f005:**
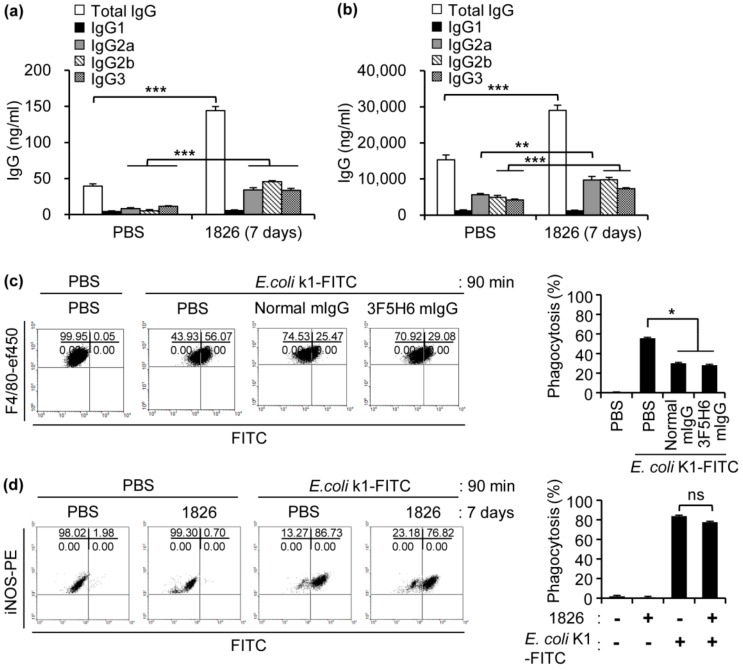
BALB/c mice were i.p. injected with PBS or CpG-DNA 1826 (2.5 mg/kg mouse). After seven days, the mice were sacrificed, and the peritoneal cavity supernatants and serum were harvested from the mice. The amounts of *E. coli* K1-reactive IgG subclasses in peritoneal cavity supernatant (**a**) and serum (**b**) were measured by ELISA using *E. coli* K1-coated plates (*n* = 3/group). 1826, CpG-DNA 1826. ** *p* < 0.005, *** *p* < 0.0005. (**c**) Effect of bacteria-reactive monoclonal antibody (3F5H6 mIgG) on phagocytosis in the mouse peritoneal cavity against *E. coli* K1 infection. FITC-labeled *E.*
*coli* K1 cells (1 × 10^6^ CFU/mL) were incubated with PBS, normal mouse IgG, or 3F5H6 mIg (10 μg/mL) for 1 h and then 200 μL of the mixture was i.p. injected into BALB/c mice (*n* = 3). After 90 min, peritoneal cells were harvested from the mice and stained with specific markers. FSC^low^SSC^high^ cells of peritoneal cells were gated, and the myeloid populations of the peritoneal cavity were gated using F4/80 subsets. FITC-labeled *E. coli* K1 uptake of macrophages was determined by flow cytometry. 1826, CpG-DNA 1826. Normal mIgG, normal mouse IgG. 3F5H6 mIgG, CpG-DNA-induced bacteria-reactive monoclonal antibody clone 3F5H6. * *p* < 0.05. (**d**) Analysis of *E. coli* K1 uptake in the mouse peritoneal cavity. BALB/c mice were i.p. injected with PBS or CpG-DNA 1826 (2.5 mg/kg mouse). After seven days, the mice were i.p. injected with PBS or FITC-labeled *E. coli* K1 (2 × 10^6^ CFU/mouse). After 90 min, peritoneal cells were harvested, counted, and stained with fluorescence-conjugated antibodies and analyzed by flow cytometry. FSC^low^SSC^high^ cells of peritoneal cells were gated, and the myeloid populations of the peritoneal cavity were gated using F4/80 and sorted into CD206 and iNOS subsets. FITC-labeled-*E. coli* K1 uptake of M1 macrophages was determined by flow cytometry. 1826, CpG-DNA 1826. ns, not significant.

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
