# Peer review of "Anti-Bacterial Effect of CpG-DNA Involves Enhancement of the Complement Systems"

_ijms, 2019, doi:10.3390/ijms20143397_

Round 1

Reviewer 1 Report

 In this study, Kim et. al. found that the enhancement of complement systems by CpG-DNA contributed to the anti-bacterial effect of CpG-DNA treatment. Generally, the figures are easy to follow and data are well organized. However, there are some concerns needing to be addressed.

1. In Fig 2c and 2d, it's better to include the bacteria E.coli K1 loading in different organs and histology analysis from the control group of PBS+ E.coli K1. 

2. It's reported the TRIM29 could be induced by dsDNA (Nat Commun. 2017, 8(1):945) and TRIM29 could negatively regulate the bacteria infection by targeting NEMO and played protective role in host defense against bacteria infection (Nat Immunol. 2016,17(12):1373-1380). Whether CpG-DNA induced the expression of TRIM29 and the enhanced TRIM29 could play protective role in host defense against E. coli infection, which should be discussed.

Author Response

1. In Fig 2c and 2d, it's better to include the bacteria E.coli K1 loading in different organs and histology analysis from the control group of PBS+ E.coli K1. 

- The data shown in Figure 2c and d are the results of 7 days after E. coli K1 infection. In the control group of PBS + E. coli K1, all the mice died in two days after infection. Therefore, it was impossible for histology analysis. In Figure 1c and d, histology analysis of the tissues in the control group was performed one day after infection.

2. It's reported the TRIM29 could be induced by dsDNA (Nat Commun. 2017, 8(1):945) and TRIM29 could negatively regulate the bacteria infection by targeting NEMO and played protective role in host defense against bacteria infection (Nat Immunol. 2016,17(12):1373-1380). Whether CpG-DNA induced the expression of TRIM29 and the enhanced TRIM29 could play protective role in host defense against E. coli infection, which should be discussed.

- We discussed this point in the Discussion as followings. Thanks for your suggestion.

“Previously, it was reported that the E3 ubiquitin ligase tripartite motif protein 29 (TRIM29) could be induced by dsDNA [35]. TRIM29 could negatively regulate the infection of gram negative bacteria Haemophilus influenza by targeting NEMO and played protective role in host defense against bacterial infection [36]. Therefore, we can consider possible involvement of TRIM29 in the protective function of CpG-DNA against E. coli K. However, it is not likely because expression of TRIM29 was not induced by CpG-DNA treatment [37].”

Reviewer 2 Report

Critiques for manuscript “Anti-bacterial effect of CpG-DNA involves enhancement of the complement systems” by Kim et al.:

The authors have reported the therapeutic effect of CpG (class B) against E. coli K1 infection in mice. Improvement in survival in these mice is paralleled by an increase in C3 complement. The data is very interesting and provides brief insights into alternative ways that CpG can act on immune system. The manuscript is written in a clear, concise manner and falls within the scope of the journal. However, where the manuscript feels lacking is in evaluating the mechanisms behind the observed therapeutic effect of CpG.

Some of my suggestions to improve the manuscript is as follows:

1.     It is interesting to see CpG treatment results in increased C3 and reduced C3a/C3b products. Does this imply a reduced cleavage of C3 to C3a/C3b or an increased induction of C3 synthesis by CpG?

2.     What are the levels of other complement components upstream of C3 in CpG-treated mice (e.g. C1q, C2, C4 etc)?

3.     A note on the specificity of C3 ELISA assay (whether it also detects C3a and C3b or not) should be mentioned by the authors.

4.     What is the source of increased C3 component after CpG treatment? Obviously, reduced C3 cleavage after K1 infection could be an explanation. However, increase in C3a/C3b is not mirrored by a decrease in C3 levels (Figure 3b) after E. coli infection.

5.     In Figure 4, CpG improves survival in mice injected with CVF and E. coli when compared to mice injected with CVF and E. coli (Figure 4d). However, CpG pre-treatment does not reduce C3 cleavage by CVF (Figure 4b). The correlation between increased C3 complement levels and survival in CpG-treated mice is not consistently noticed between experiments, implying that it is not the major determinant.

6.     CpG 1826 is a class B CpG DNA and mainly activates B cells. Thus, the effect of B cells on the observed CpG effect becomes important. What is the role of B cells in increasing C3 complement? Does CpG results in increased B cell activation in these mice?

7.     Do mice that recovered from K1 infection (Figure 2) show presence of neutralizing anti-K1 antibodies? What is the effect of CpG on anti-bacterial IgG production?

8.     Can the authors comment on any additional pathways (other immune cells) involved in CpG-induced recovery after E. coli infection in mice?

9.     Mouse macrophages seem to express TLR9 (An et al., 2002, Immunol Letters). Do mice pre-treated with CpG show increased macrophage activity (e.g. phagocytosis) against E. coli infection, especially in recovered mice?

Overall, the authors have reported an interesting observation linking CpG, complement and survival against E. coli infection in mice, which is novel. However, there is a lack in experiments to understand the mechanism. Additional studies, as listed above, might help provide some mechanistic insights into this role.

Author Response

1. It is interesting to see CpG treatment results in increased C3 and reduced C3a/C3b products. Does this imply a reduced cleavage of C3 to C3a/C3b or an increased induction of C3 synthesis by CpG?

- As CpG-DNA alone didn’t increase the level of C3 at the time point (8 days) we examined, increased C3 level may be related with a reduced cleavage of C3 to C3a/C3b owing to the CpG-DNA pretreatment. We added this in the result section explaining Figure 3b.

 We also added following information in the Discussion for better understanding.

- In this study, we used phosphorothioate-modified CpG-DNA 1826. It was previously reported that phosphorothioate-modified oligonucleotides reduced levels of factor H, an inhibitor of the alternative pathway (AP) convertase, and affect alternative pathway leading to complement activation [29, 30]. It was also reported that CpG-DNA increases the levels of C3a and C5a via classical and alternative pathways [7]. Here, we found that injection of CpG-DNA1826 into mice induced transient increase of C3 followed by later and transient increase of C3a and C3b.

2.     What are the levels of other complement components upstream of C3 in CpG-treated mice (e.g. C1q, C2, C4 etc)?

- There is a report that ssDNA and dsDNA can bind with C1q and affect the classical pathway [34]. We wrote this information in the Discussion as following.

Considering that there is a report that ssDNA and dsDNA can bind with C1q and affect the classical pathway [34], function of other complements such as C1q, C2, and C4 can be involved. In this study, we didn’t check other complements, but focused on the fact that CpG-DNA is involved in complement activation and contributes to the suppression of E. coli K infection. Detailed effects of CpG-DNA on complement system are to be determined in the future.”

3.     A note on the specificity of C3 ELISA assay (whether it also detects C3a and C3b or not) should be mentioned by the authors.

- To check the specificity of the ELISA assay kits, we performed experiment using each standard and confirmed that each ELISA kit is specific to C3, C3a, and C3b, respectively. We added this information in the Materials and Methods as followings.

“The Mouse C3 ELISA Kits didn’t have cross-reactivity to C3a and C3b standard. The Mouse C3a and Mouse C3b ELISA Kits didn’t react with C3 standard, either.”

4.     What is the source of increased C3 component after CpG treatment? Obviously, reduced C3 cleavage after K1 infection could be an explanation. However, increase in C3a/C3b is not mirrored by a decrease in C3 levels (Figure 3b) after E. coli infection.

- E. coli K1 infection can activate complement system and increase C3a and C3b through classical pathway, alternative pathway, and lectin pathway. Therefore, the reason why the amount of C3 didn’t reduce greatly is presumed to be the induction of C3 in response to E. coli K1 infection.

In the mice pre-treated with CpG-DNA prior to E. coli K1 infection, the C3 levels were significantly increased, but the levels of C3a and C3b were similar to those of the control mice (Figure 3b). Therefore, it is likely that the increase of C3 level is related a reduced cleavage of C3 to C3a/C3b owing to the CpG-DNA treatment. It suggests that the regulation of these complement levels might contribute to the survival of the mice against E. coli K1 infection.

We added this information in the Results section explaining Figure 3b.

5.     In Figure 4, CpG improves survival in mice injected with CVF and E. coli when compared to mice injected with CVF and E. coli (Figure 4d). However, CpG pre-treatment does not reduce C3 cleavage by CVF (Figure 4b). The correlation between increased C3 complement levels and survival in CpG-treated mice is not consistently noticed between experiments, implying that it is not the major determinant.

In Fig. 4d, all the mice pretreated with CpG-DNA before E. coli K1 infection survived (100%), but the survival was greatly reduced by CVF treatment (20% survival). Therefore, the result show that complement 3 is required for survival.

As CVF depletes all C3 almost completely in our experimental setting, we can confirm that the function of complement is essential in the clearance of E. coli K1. If we perform experiments using various concentrations of CVF, we possibly see the differential effect owing to the increased C3 level induced by CpG. Please consider that this optimization is a big burden to us.

6.     CpG 1826 is a class B CpG DNA and mainly activates B cells. Thus, the effect of B cells on the observed CpG effect becomes important. What is the role of B cells in increasing C3 complement? Does CpG results in increased B cell activation in these mice?

- CpG-DNA activates the complement system in the whole blood loop system. Production of complement components are produced by hepatocytes and endothelial and epithelial cells and by immune cells such as monocytes, macrophages, and dendritic cells. Surely antigen-reactive antibodies produced by B cells can activate complement system, however we don’t know what the role of B cells is in increasing the amount of C3 complement.

- Clear thing is that CpG results in increased B cell activation and induces bacteria-reactive antibodies in the mice. We added the results as Fig. 5 and added the information in the Discussion.

“In addition, we found that CpG-DNA activated peritoneal B cells to produce bacteria-reactive antibodies and the antibodies increased phagocytosis of MRSA in macrophages. In this study, we added results showing that the antibodies are reactive E. coli K1 (Figure 5a).” 

7.     Do mice that recovered from K1 infection (Figure 2) show presence of neutralizing anti-K1 antibodies? What is the effect of CpG on anti-bacterial IgG production?

- We added following information in the results (Fig. 5) and discussion.

“We previously reported that CpG-DNA stimulates peritoneal B cells, increases production of bacteria-reactive antibodies, and that the antibodies increase phagocytosis of MRSA [17]. In this study, we additionally observed antibodies reactive to E. coli K1 in the peritoneal cavity and serum of CpG-DNA-stimulated mice (Figure 5a, 5b). However, the antibodies can’t enhance phagocytic activity of macrophages against E. coli infection (Figure 5 c and d). We don’t know whether the antibodies have neutralizing effect against E. coli K1 or not currently.”

8.     Can the authors comment on any additional pathways (other immune cells) involved in CpG-induced recovery after E. coli infection in mice?

- We specified CpG-DNA responsive immune cell populations such as B cells, monocytes, macrophages, and dendritic cells in the Discussion. As CpG-DNA overall stimulate these cells, CpG-DNA activates innate immunity and adapted immunity and may contributes to the recovery after E. coli infection in mice. 

“Considering that several types of immune cell populations such as B cells, monocytes, macrophages, and dendritic cells were activated after CpG-DNA administration [17], we suppose that these CpG-DNA-activated immune cells and immunoregulators including complement components are involved in the enhanced survival of the mice and bacterial clearance in the examined organs after E. coli K1 infection.”

- We described function of CpG-DNA involved in defense to bacterial infection in the introduction. We believe that such mechanism can work for the recovery after E. coli infection.

“CpG-DNA treatment stimulates the innate immunity including the activation of dendritic cells and the induction of oxidative stress against the infections of several bacteria such as Listeria monocytogenes [12], Staphylococcus aureus (S. aureus) [13], Salmonella typhimurium [14] and Klebsiella pneumonia [15]. Furthermore, systemic administration of CpG-DNA prevents intracerebral Escherichia coli (E. coli) K1 infection by increasing the monocytes in brain and spleen of neutropenic mouse [16]. In recent studies, we proved the protective effects of the antibodies produced by CpG-DNA administration against MRSA, one of the Gram-positive bacteria, infection [17, 18]”

- While the peritoneal cavity macrophage of BALB/c mice showed phagocytosis of MRSA at 20% efficiency (ref), phagocytosis of E. coli K1 was found at about 60% efficiency in the absence of CpG-DNA treatment suggesting more efficient removal of E. coli K1 in the mice.

We added this information in the Discussion.

9.     Mouse macrophages seem to express TLR9 (An et al., 2002, Immunol Letters). Do mice pre-treated with CpG show increased macrophage activity (e.g. phagocytosis) against E. coli infection, especially in recovered mice?

- We previously studied the effect of CpG-DNA on the phagocytic activity of macrophages [17]. Stimulation of peritoneal B cells with CpG-DNA induced bacteria-reactive antibodies in vitro and the antibodies enhanced phagocytic activity of peritoneal cells when used MRSA as an infecting bacterium.

Differently from MRSA, injection of CpG-DNA didn’t increase uptake of E. coli K1 in the peritoneal macrophages (Figure 5). Probably, there are some differences in the immune reactions in response to Gram positive bacteria versus Gram-negative bacteria. We don’t know exact reason at this time and it has to be further investigated.

We added this information in the Discussion.

“In this study, we additionally observed antibodies reactive to E. coli K1 in the peritoneal cavity and serum of CpG-DNA-stimulated mice (Figure 5a, 5b). Differently from MRSA, CpG-DNA-induced bacteria-reactive monoclonal antibody (3F5H6 mIgG) didn’t increase but decrease phagocytosis of E. coli K1 (Figure 5c). This suggest a possibility that 3F5H6 mIgG binds to E. coli K1 and inhibit the uptake of macrophage. Injection of CpG-DNA didn’t increase uptake of E. coli K1 in the peritoneal macrophages, either (Figure 5c). While the peritoneal cavity macrophage of BALB/c mice showed phagocytosis of MRSA at 20% efficiency [17], phagocytosis of E. coli K1 was found at about 60% efficiency in the absence of CpG-DNA treatment suggesting the presence of mechanism inducing more efficient removal of gram negative bacteria in the mice.”

Overall, the authors have reported an interesting observation linking CpG, complement and survival against E. coli infection in mice, which is novel. However, there is a lack in experiments to understand the mechanism. Additional studies, as listed above, might help provide some mechanistic insights into this role.

- Thanks for your thoughtful suggestion. We previously investigated effects of CpG-DNA on immune cells in a variety of aspects and here we are focusing on the effects on E. coli K1 infection. We described previous findings, supplemented more results, and tried to supplement discussion as much as we can. Thanks.

Round 2

Reviewer 2 Report

The questions raised by the reviewer(s) has been answered and the data requested were provided. No additional changes are needed.